# Arbicon-Net: Arbitrary Continuous Geometric Transformation Networks for Image Registration

**Jianchun Chen** *
NYU Multimedia and Visual Computing Lab
New York University
Brooklyn, NY 11201
jc7009@nyu.edu

**Lingjing Wang** *
NYU Multimedia and Visual Computing Lab
New York University
Brooklyn, NY 11201
lw1474@nyu.edu

**Xiang Li**
NYU Multimedia and Visual Computing Lab
New York University
Brooklyn, NY 11201
xl845@nyu.edu

**Yi Fang** †
NYU Multimedia and Visual Computing Lab
New York University Abu Dhabi
Abu Dhabi, UAE
yfang@nyu.edu

## Abstract

This paper concerns the undetermined problem of estimating geometric transformation between image pairs. Recent methods introduce deep neural networks to predict the controlling parameters of hand-crafted geometric transformation models (e.g. thin-plate spline) for image registration and matching. However the low-dimension parametric models are incapable of estimating a highly complex geometric transform with limited flexibility to model the actual geometric deformation from image pairs. To address this issue, we present an end-to-end trainable deep neural networks, named Arbitrary Continuous Geometric Transformation Networks (Arbicon-Net), to directly predict the dense displacement field for pairwise image alignment. Arbicon-Net is generalized from training data to predict the desired arbitrary continuous geometric transformation in a data-driven manner for unseen new pair of images. Particularly, without imposing penalization terms, the predicted displacement vector function is proven to be spatially continuous and smooth. To verify the performance of Arbicon-Net, we conducted semantic alignment tests over both synthetic and real image dataset with various experimental settings. The results demonstrate that Arbicon-Net outperforms the previous image alignment techniques in identifying the image correspondences.

## 1 Introduction

Image registration plays a fundamental role in many computer vision applications such as medical image processing [1], camera pose estimation [2], visual tracking [3]. Fig.1 shows the image registration process, which includes geometric transformation estimation and image warping. To formulate the problem of image registration, traditional methods often approach the task in two steps: 1) they firstly compute the hand-crafted image features such as SIFT and HOG [4, 5] to capture pixel-level descriptions, 2) and then iteratively search the optimal geometric transformation model to register a pair of images, driven by minimizing an alignment loss function. The alignment loss is usually pre-defined as a certain type of similarity metric (e.g. correlation scores) between two

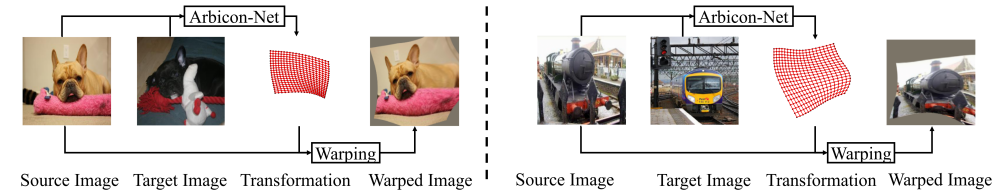

| Source Image | Target Image | Transformation | Warped Image | | Source Image | Target Image | Transformation | Warped Image |

Figure 1: Illustration of Arbicon-Net for image alignment.

sets of image feature descriptors. Previous efforts [6, 7, 8] have achieved great success in image registration through the development of a variety of image feature descriptors and optimization algorithms as summarized in [9]. However, they often face challenges posed by various deteriorated image conditions such as 1) the dramatic image appearance variation (i.e. texture, color, lighting changes and so on) between image pairs, and 2) the significant geometric structural variation between image pairs.

The recent success of deep neural network motivates researchers [10, 11, 12, 13] to develop deep learning techniques to combine both two steps into an end-to-end trainable network, which aims to learn a pre-defined geometric model (i.e. affine or thin-plate spline) through the regression process supervised by minimizing the image matching loss. With the generalization from training data, those methods are able to predict real-time image matching that is robust to various deteriorated image conditions. However, it is suggested by the authors [14] that pre-defined geometric transformation models only represent a set of low dimension transformations which prevents these methods from predicting complex geometric transformations for high-quality image registration. Moreover, the transformations described by hand-crafted geometric models might not reveal the actual transformation required for image alignment, which leads to a sub-optimal estimation of desired geometric transformations.

Some methods [14, 15, 16] tackle this problem by directly estimating semantic flow from pixel-level features. These methods are more flexible to transfer the keypoints of images to semantically correlated positions. However, since the flow field is estimated entirely by local features without integrating global motion, local points are unable to move coherently, which consequently generates distorted unrealistic images. In real-world applications (e.g. [1]), these flow based methods require explicitly imposed penalization to constrain the smoothness of flow field.

To address the above mentioned issues, we propose to develop a novel geometric transformation network, named arbitrary continuous geometric transformation networks (Arbicon-Net), to directly predict the dense displacement field that is not formulated by pre-defined hand-crafted geometric models. Compared with geometric model based approaches, Arbicon-Net uses deep neural network to model geometric transformations to accommodate arbitrary complex transformations required for the registration of image pairs. Compared with semantic flow based methods, Arbicon-Net features an attractive property, which predicts a smooth displacement field. As shown in Fig.2, we design an Arbicon-Net to simultaneously train three major modules, namely front-end geometric feature extractor module, transformation descriptor encoder module and displacement field predictor module, in an end-to-end fashion. The Arbicon-Net firstly extracts dense feature maps from input image pairs and encodes the discriminative local feature correlation into a transformation descriptor. The following predictor module uses the transformation descriptor to decode displacement field for image registration.

**Contributions.** We have three main contributions in this paper. First, we design a novel Arbicon-Net, which uses deep neural networks to predict dense displacement field to accommodate the arbitrary geometric transformations according to the actual requirement for image registration. This addresses the critical issue that the actual desired geometric transformation does not match with the one that can be provided by pre-defined geometric model. Second, we prove that the Arbicon-Net is guaranteed to generate spatially continuous and smooth displacement field without imposing additional penalization term as a smoothness constraint. Finally, we show that our proposed Arbicon-Net achieved superior performance against hand-crafted geometric transformation models with both strong and weak supervision.

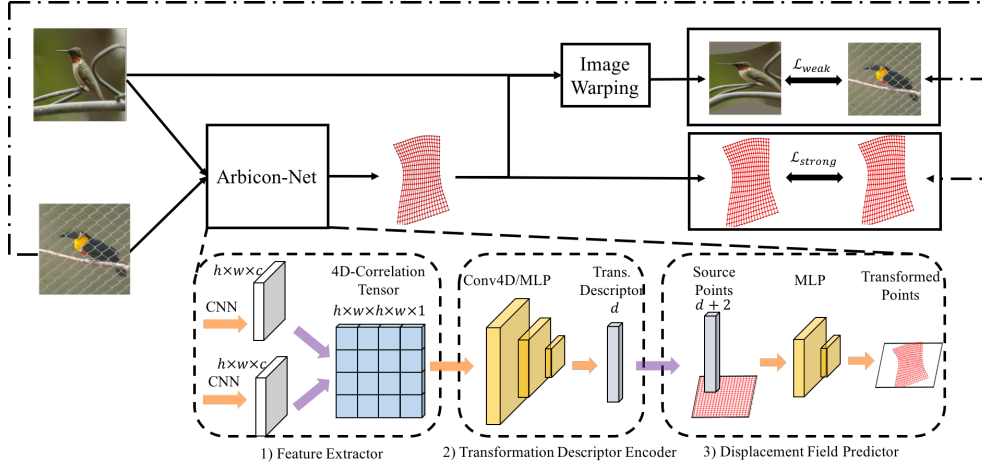

Figure 2: Main Pipeline. Our proposed end-to-end trainable Arbicon-Net has three main components. 1) Geometric Feature Extractor Module; 2) Transformation Descriptor Encoder Module; 3) Displacement Field Predictor Module.

## 2 Related Works

**Image Registration.** Image registration is defined as a process to determine a smooth geometric transformation between input image pairs, especially for 2D/3D medical images. Existing methods search optimal geometric transformation by iteratively minimizing alignment loss, which is typically defined by the feature similarity or hierarchically defined intensity pattern. To achieve a high-quality image registration, researchers [17, 1, 9] have explored diverse geometric transformation models, image similarity metrics and searching algorithms.

**Non-learning based Image Correspondence Matching.** The classic image correspondence matching pipeline [6, 7, 8] starts by detecting key points via hand-crafted pixel-level feature descriptors [4, 5, 18], followed by feature matching strategies to determine the optimal point correspondence [19, 5]. Following researches have developed various hand-crafted algorithms [19, 20] to remove incorrect matches by searching global transformation or utilizing neighbor information. While these methods are limited in matching speed and matching performance, they have far-reaching impact on computer vision society by imposing a standard pipeline and introducing geometric transformation estimation as a mainstream approach for image matching problem.

**Learning based Image Correspondence Matching.** Inspired by the success of deep neural network, pioneer works [21, 22, 23] propose to use pre-trained convolutional neural networks (CNNs) instead of hand-crafted ones to extract discriminative pixel-wise feature descriptors. Following researches develop learnable feature extraction layer [24, 25] and learnable feature matching layer [26, 27] with differentiable image alignment loss. Han et al. [28] introduce a fully learnable image correspondence matching strategy over region proposals. However, this method is not in an end-to-end trainable fashion.

More recently, researchers [10, 11, 12, 13] propose end-to-end trainable network architectures for image correspondence estimation. Specifically, these methods define a regression network to predict the parameter of specific geometric transformation models (i.e. thin-plate spline, affine). But they are limited by the use of low dimension geometric models and consequently less capable of performing fine-grained image geometric transformation. Other researchers either recurrently regress pixel-level flow field [15, 16] to approximate fine-grained image transformation or determine flow field by neighbourhood consensus assignment [14]. However, as we stated above, they don't take the smoothness of displacement field into account.

## 3 Approach

### 3.1 Geometric Feature Extractor

Following the common image matching paradigms [11], our Arbicon-Net starts with extracting geometric features from input image pair $I_A, I_B$. We firstly leverage a share-weighted CNN to generate a representative feature map $F \in \mathbb{R}^{h \times w \times c}$ for each input image, where at each location the feature vector $f_{ij} \in \mathbb{R}^c$ represents local semantic information.

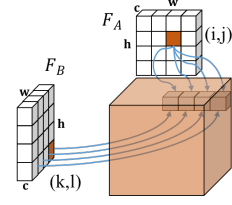

In order to estimate the geometric transformation of given image pairs, we establish the local feature correlations between two feature maps by using the normalized cosine similarity. For each local descriptor from $F_A$, we compute its similarity score with all local descriptors in $F_B$ to form a 4-D correlation tensor $\mathbf{S} \in \mathbb{R}^{h \times w \times h \times w}$ as shown in Fig.3. Each element $s_{ijkl} \in \mathbf{S}$ is computed as,

Figure 3: Feature Correlation.

$$s_{ijkl} = \frac{\langle f_{ij}^A, f_{kl}^B \rangle}{||f_{ij}^A||_2 ||f_{kl}^B||_2} \qquad (1)$$

where $\langle \cdot \rangle$ denotes the inner product of two vectors, the denominator acts a normalization term to further amplify confident matching and reduce ambiguity matching.

## 3.2 Transformation Descriptor Encoder

To learn a more discriminative feature correlation, we leverage 4-D convolutional neural networks (CNNs) to refine correlation tensor $S$ by using neighbor information [14]. The 4-D convolution layers integrate additional neighborhood information compared with regular 2-D CNNs. Since the order of input image pairs $(I_A, I_B)$ or $(I_B, I_A)$ do not influence the result of local feature correlation, the convolution operation is symmetrically applied, formulated as,

$$S^C = Conv(S) + (Conv(S^T))^T \qquad (2)$$

, where the transpose of $S$ is computed according to $s_{ijkl}^T = s_{klij}$. Moreover, we normalize the learned 4-D correlation tensor $s$ by Eq.3, where $\phi_{pq}^A = \{s_{pq11}^c, ..., s_{pqhw}^c\}$ and $\phi_{pq}^B = \{s_{11pq}^c, ..., s_{hwpq}^c\}$. This normalization encourages the bilateral confidence of correlated pairs for from source image and target image.

$$\hat{s}_{ijkl} = \frac{s_{ijkl}^c}{max(\phi_{ij}^A)} \frac{s_{ijkl}^c}{max(\phi_{kl}^B)} s_{ijkl}^c \qquad (3)$$

Since our goal is to find a global transformation, we use a Multi-Layer Perceptron to encode learned 4-D tensor $\hat{\mathbf{S}}$ into a transformation descriptor $d_{AB} \in \mathbb{R}^m$ that represents the overall image correspondence information, as shown in Eq.4. For global geometric transformation learning, the image correspondence information describes a geometric transformation that optimally aligns corresponding points on two images.

$$d_{AB} = MLP(\hat{S}) \qquad (4)$$

## 3.3 Displacement Field Predictor

In general, the geometric transformation $\mathcal{T}$ for each point $x$ in a point set $\mathbf{X} \subset \mathbb{R}^2$ can be defined as:

$$\mathcal{T}(x, v) = x + v(x) \qquad (5)$$

, where $v : \mathbb{R}^2 \to \mathbb{R}^2$ is a "point displacement" function.

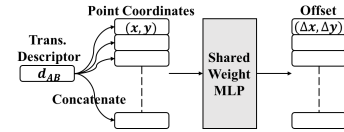

The image registration task can be formulated as a process of determining the displacement function $v$. It is necessary for function $v$ to be a continuous and smooth function according to the Motion Coherent Theory (MCT) [29]. Fortunately, by leveraging deep neural network architecture, we can construct the suitable displacement function $v$ which satisfies the continuous and smooth characteristics.

Figure 4: Displacement Field Predictor Module.

As illustrated in Fig.4, given $n$ 2-D points in the source image plane, we duplicate the transformation descriptor $d_{AB}$ for $n$ times.

Each point is concatenated with the $m$-D global descriptor $d_{AB}$. We further construct a Displacement Field Predictor network with four successive MLPs to decode the concatenated $(m+2)$-D vector into 2-D displacement vector. We defined this neural network structure in Eq.6 as $\mathcal{F}(\cdot) : \mathbb{R}^{m+2} \to \mathbb{R}^2$, formulated as,

$$v(x) = \mathcal{F}([x, d_{AB}]) \tag{6}$$

,where $[\cdot]$ indicates concatenation operation.

Furthermore, we briefly prove the continuity and smoothness of our displacement field predictor $\mathcal{F}$ to be used as our deep learning-based solution for the displacement function $v$.

**Continuity**. Since both MLP and activation function $\sigma$ are continuous, the continuity of Displacement Field Predictor network can be trivially proven as a composite of continuous functions. Since $d_{AB}$ is concatenated to each point in $\mathbf{X}$, this concatenation operation does not change the continuity for displacement function $v(\cdot) = \mathcal{F}([\cdot, d_{AB}])$. In contrast, commonly used learning paradigms [1, 15, 16], which directly map high dimension feature space to 2-D displacement field, output a set of discrete displacement vectors, while the displacements of other points need to be further interpolated.

**Smoothness.** After choosing a smooth function SoftPlus [30] as the activation function in our Displacement Field Predictor network, it becomes trivial to estimate its complexity and smoothness since the displacement function is a composite of a number of smooth functions (MLP and SoftPlus). In practice, Regularization Theory (RT) [31] uses the oscillatory behavior of a function to further measure the smoothness of displacement function. The oscillatory behavior is measured by Reproducing Kernel Hilbert Space (RKHS) [31, 32] in Eq.7.

$$||v||_{\mathbb{H}^m}^2 = \int_{\mathbb{R}^D} \frac{|\tilde{v}(s)|^2}{\tilde{g}(s)} ds \tag{7}$$

, where $\tilde{v}$ is the Fourier transform of the displacement function $v$ and $\tilde{g}$ is a low-pass filter. In other words, a smoother displacement function has considerably less energy in high frequency domain. We generally express models that regress pixel-level displacement vector, including Arbicon-Net and RTNs [16], as a composite function $v$ in Eq.8.

$$v(x) = \mathcal{F}(\mathcal{G}(x)) : \quad \mathbb{R}^2 \to \mathbb{R}^2 \tag{8}$$

Function $\mathcal{G}$ and $\mathcal{F}$ denotes the point feature encoding network and point displacement vector regression network respectively. Specifically, we have $\mathcal{G}(x) = [x, d_{AB}]$ in Arbicon-Net. In contrast, in RTNs $\mathcal{G}$ generates a high-dimensional feature map by sequential CNNs. Therefore, the $\mathcal{G}$ in RTNs is generally considered as a sparse and oscillate function, especially when the dimension of feature vector (output of function $\mathcal{G}$) is high, which causes the widely known "curse of dimensionality" problem. In this section, we assume that two models have same regression network $\mathcal{F}$ and the input of function $\mathcal{F}$ are normalized to a same scale.

According to [33], the Fourier transform of the composite function $v$ has essentially maximum frequency $u_v$ as

$$u_v = u_{\mathcal{F}} \max_x |\mathcal{G}'(x)| \tag{9}$$

, where $u_{\mathcal{F}}$ is the maximum frequency of $\mathcal{F}$, which is independent of $\mathcal{G}$. Assume that the outputs of different function $\mathcal{G}$ are in a same scale, the oscillate function $\mathcal{G}$ tends to have a larger maximum value of $|\mathcal{G}'(x)|$ compared with linear function in Arbicon-Net. As a result, our composite function has a smaller $u_v$, which is likely to have lower energy in high frequency domain, which further guarantees a smoother displacement function.

Based on our proposed paradigm, we further constrain the smoothness of function $\mathcal{F}$. Fortunately, given the popularity of deep learning models, the recent research community has been proposing regularization strategies, which naturally help our Displacement Field Predictor network to reduce the risk of the oscillatory of displacement function. One simple solution is to design a proper network size. In section 4.3, we provide empirical results to validate the smoothness of our estimated displacement function compared with non-rigid geometric transformation models.

## 3.4 Loss functions

As shown in the right box of Fig.2, our designed method is designed to learn geometric transformation under either strong supervision or weak supervision.

For strongly-supervised loss, we have point correspondence information of $x \in \mathbf{X}$ from source plane and $y \in \mathbf{Y}$ from target plane. $L_{strong}$ directly minimizes the pairwise L2 distance between corresponding points in transformed image plane and target image plane, as shown in Eq.10.

$$L_{strong} = \frac{1}{N} \sum_{i=1}^{N} ||\mathcal{T}(x_i) - y_i||_2^2 \tag{10}$$

For weakly-supervised loss, we maximize the inner product of corresponding location in transformed source feature map $\mathcal{T}(F_A)$ and target feature map $F_B$ following the paradigm described in [10]. Let $\mathcal{T}(I_A)$ and $I_B$ to be matched, we have $\mathcal{T}(I_A)^{ij}$ and $I_B^{ij}$ to be semantically matched, thus $\left\langle \mathcal{T}(f_A)^{ij} \cdot f_B^{ij} \right\rangle$ to be maximum. We implement the loss function described in Eq.11.

$$L_{weak} = -\sum_{i,j,k,l} s_{ijkl} \mathbf{1}_{d(\mathcal{T}(i,j),(k,l))<t} \tag{11}$$

, where $\mathbf{1}(\cdot)$ is the indicator function, $d(\cdot)$ denotes L1 distance.

Since our proposed network is end-to-end trainable, the Geometric Transformation Network is optimized together with other components. It deserves noting that, since our method learns the geometric transformation from training dataset, it is more robust to train or fine-tune our network on real image dataset than transfer a pre-trained network from a one-fold synthesized dataset.

## 4 Experiments

In this section, we carried out a set of tests under different experimental settings to validate the performance of our proposed Arbicon-Net for its capability of estimating the geometric transformation for image dense correspondence in semantic alignment.

In section 4.1, we describe the implementation details of our Arbicon-Net to be tested in our experiments. In section 4.2, we discuss the details of the experimental dataset preparation, evaluation metric for experimental results and baseline models for experimental performance comparison. In section 4.3, we validate the performance of Arbicon-Net for the estimation of the geometric transformation for real image pairs with weakly supervised training. In section 4.4 and 4.5, we demonstrate the performance of Arbicon-Net for the estimation of parametric and non-parametric geometric transformation respectively, and compare to the state-of-the-art techniques.

### 4.1 Implementation Details

As shown in Fig.2, the Arbicon-Net is implemented based on deep neural networks with the following architecture configuration. Arbicon-Net starts with the use of ResNet [34] (before *conv4-23*) with weight pre-trained on ImageNet for local feature extraction, then followed by three 4-D convolution kernels in Section 3.2 which are of size $(3, 3, 3)$ with channels $(10, 10, 1)$ respectively, and end with four MLPs configured with the size $(256, 256, 64, 2)$. In Arbicon-Net, we set the dimension of translation descriptor $d_{AB}$ at 256 as shown in Fig.4. For weakly supervised Arbicon-Net (refer to Section 4.3), we first train our model on synthesize dataset and fine-tune our model on training set of PF-PASCAL using loss function in Eq.11. We refer [10] for detailed loss function and training setup. For supervised Arbicon-Net, we use Adam optimizer for training with learning rate 0.001. The network is implemented by PyTorch framework and ran on an Nvidia GTX 1080Ti GPU.

### 4.2 Dataset, Metric and Baseline

**Dataset.** In our experiment, three image datasets, Pascal VOC dataset [35], PF-Pascal dataset [36] and Proposal Flow dataset [37] are used to prepare both synthesized and real image dataset for the various tests. Pascal VOC dataset [35] contains 28,952 images. PF-Pascal contains 1351 semantically aligned image pairs from 20 semantic category of Pascal VOC dataset with a 7:3:3 training/validation/testing split. Proposal Flow dataset [37] with 900 image pairs from 5 categories. We randomly split the Proposal Flow dataset into 3 folds for k-fold validation in the test. The image pair from PF-Pascal and Proposal Flow datasets is annotated with correspondences that could be used as ground truth for image matching performance evaluation. In order to prepare the synthesized image data, we imposed

| Methods | PCK(%) |
|---|---|
| HOG+PF-LOM [36] | 62.5 |
| SCNet-AG+ [28] | 72.2 |
| CNNGeo [11] | 71.9 |
| A2Net [13] | 70.9 |
| WeakAlign [10] | 75.8 |
| WeakAlign-4D | 76.5 |
| Arbicon-Net | **77.3** |

Table 1: Quantitative results on PF-Pascal [36] dataset with weakly supervise training.

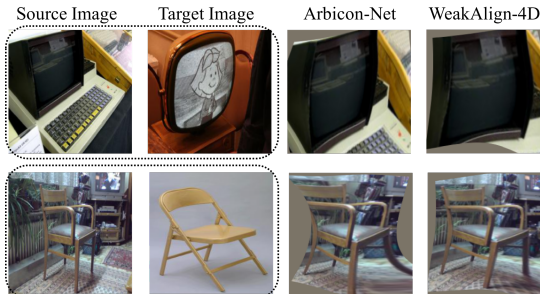

Figure 5: Qualitative results on PF-Pascal [36] dataset with weakly supervise training.

| Methods | MSE |
|---|---|
| CNNGeo-4D | 0.0037 |
| Arbicon-Net | **0.0002** |

Table 2: Quantitative result on synthesized dataset with diffeomorphic non-linear transformation.

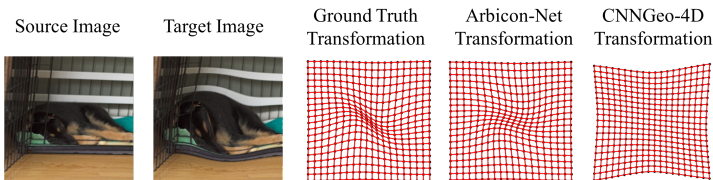

Figure 6: Qualitative comparison on synthesized dataset with diffeomorphic non-linear transformation.

two different types of parametric transformations, Thin plate spline (TPS) and Diffeomorphic non-linear transformation, onto the images in Pascal VOC dataset.

**Evaluation Metric.** We use standard evaluation protocol, the average probability of correct keypoint (PCK) for the evaluation of image matching in [38]. PCK classifies keypoint as correct match if the distance between transformed source keypoint and corresponding target keypoint is within threshold $\alpha = 0.1$ of the image size [10, 28]. For the tests on the synthesized dataset, we directly use Mean Square Error between transformed source points and corresponding target points for evaluation, as explained in Eq.10. Additionally, we measure the smoothness of estimated displacement field by the the second order derivatives of displacement field denoted as $E_{smooth}$ following [39].

**Baseline Models.** In the experiments, Arbicon-Net is compared with representative methods including HOG+PF-LOM [36], SCNet [28], CNNGeo [11], A2Net [13], WeakAlign [10]. We do not conduct the experiment for the comparison between Arbicon-Net and other semantic flow methods that directly identify sparse correspondence point without fully aligning two images. In addition, since the 4-D convolution module (refer to Section 3.2) is not proposed by us, for the fair comparison, we replace the 2-D convolution *Regression Network* module with 4-D convolution module in network structures of WeakAlign and CNNGeo, and obtain two new networks, namely WeakAlign-4D and CNNGeo-4D. We compared Arbicon-Net to both WeakAlign-4D and CNNGeo-4D in the experiments.

## 4.3   Arbicon-Net with Weakly Supervised Training

**Experiment Setting:** In this test, we conduct experiments to test the performance of Arbicon-Net for the estimation of geometric transformation for image matching without using annotated correspondence for training. In the weakly supervised setting, the Arbicon-Net and baseline models (WeakAlign and WeakAlign-4D) are firstly pre-trained on the synthesized image dataset by TPS transformation, and then followed by fine-tuning with weakly-supervised loss (refer to Eq.11). The image pairs from PF-Pascal with a certain split for are used for the training and testing of the models.

**Results Analysis:** Table 1 compares PCK scores for Arbicon-Net and baseline models for the test result on PF-Pascal dataset. The comparison result indicates that Arbicon-Net outperforms all baseline methods. To better illustrate the comparison results, we further show two pair images before and after image registration in Fig.5. The first two columns show the source and target images, the third column illustrates the transformed source image by Arbicon-Net and the fourth column illustrates the transformed source image by WeakAlign-4D. As we can see from the Figure, compared to baseline

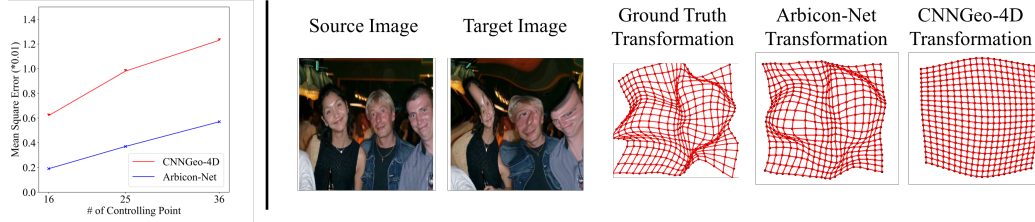

Figure 7: (a). Quantitative result on synthesized dataset with different complexity TPS transformation. (b). Qualitative result on synthesized dataset with $6 \times 6$ controlling points TPS transformation in the right side.

| Methods | PCK(%) | $E_{smooth}$ |
|---|---|---|
| CNNGeo (aff.) | 73.5 ±0.9 | 0.000 |
| CNNGeo (TPS) | 78.2 ±0.7 | 0.013 |
| CNNGeo (aff.+TPS) | 78.5 ±0.9 | 0.016 |
| Arbicon-Net | **84.3 ±0.6** | 0.008 |

Table 3: Quantitative comparisons on Proposal Flow [37] dataset (Non-parametric transformation) with strong supervision.

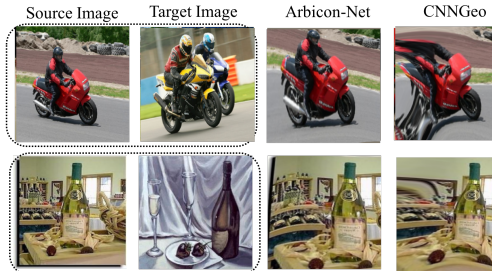

Figure 8: Qualitative comparisons on Proposal Flow [37] dataset with strong supervision.

method WeakAlign-4D, Arbicon-Net is able to predict a fine-grained geometric transformation with coherent flow motion and preservation of local geometric structural details.

In this test, Arbicon-Net and baseline models are pre-trained with TPS-based synthesized image data, and followed by fine-tuning with weakly-supervised loss. A small learning rate is posed to prevent model degeneration. In this way, the Arbicon-Net are trained with bias tendency to predict the TPS-like geometric transformation for image pair. Therefore, we design the following two additional tests for the estimation of parametric and non-parametric transformation with supervised training settings.

### 4.4 Estimation of Parametric Transformation

**Experiment Setting:** Parametric transformation means a set of geometric transformation that can be expressed by functions of controlling parameters. In this experiment, we prepare the synthesized image data with two types of parametric transformations: TPS and diffeomorphic non-linear transformation [17]. Then, we evaluate the performance of Arbicon-Net in estimating those parametric transformation after training and compare to CNNGeo-4D which uses $3 \times 3$ controlling points TPS as its geometric model. We are particularly interested in studying how Arbicon-Net addresses the critical issue that the actual desired geometric transformation does not match with the one which can be provided by pre-defined geometric model. To this end, we prepared two tests: 1) we synthesize different levels of transformed images on Pascal VOC by varying the number of TPS controlling (increasing from $4 \times 4$ to $6 \times 6$) while the CNNGeo-4D remains the same TPS parameter setting, 2) we implement a diffeomorphic non-linear transformation (a non-TPS type transformation) to synthetic data. For simplicity, we ignore the translation term to only simulate a simple 4 DoF local affine transformation around the center of image.

**Results Analysis:** The experimental results of the first test are illustrated in Fig.7. As shown in Fig.7, CNNGeo-4D (red curve) has consistently about 1.5 times higher MSE value than that of Arbicon-Net (blue curve). When testing on synthetic data simulated by $6 \times 6$ controlling point TPS, we visualize the estimated transformation in red mesh to compare the performance between Arbicon-Net and CNNGeo-4D. As we can see, Arbicon-Net clearly predicts a much more accurate displacement field with more refined and detailed local deformation, in contrast to CNNGeo-4D. The performance

deficiency by CNNGeo-4D highlights that mis-match between actual desired transformation and the one provided by a pre-defined geometric model could dramatically impact on the estimation of geometric transformation. The experimental results of the second test are shown in Table 2 and Fig.6. As we can see from the Table, Arbicon-Net clearly outperforms the CNNGeo-4D with a great margin. In the Fig.6, CNNGeo-4D failed to estimate the underlying geometric transformation while our Arbicon-Net can predict a high-quality one. This observation suggests that CNNGeo-4D limits its ability to estimate a TPS-based geometric transformation while Arbicon-Net is able to estimate an arbitrary geometric transformation for a given image pair, which is more suitable for estimating undetermined transformation.

### 4.5 Estimation of Non-parametric Transformation

**Experiment Setting:** In this experiment, we investigate the performance of Arbicon-Net in estimation of non-parametric transformation. We consider the transformation between real image pairs in Proposal Flow dataset to be non-parametric transformation, as there is no specific parametric geometric model that can be generalized to describe the transformation for any real image pair. When testing, Arbicon-Net and CNNGeo are firstly trained with real image pair supervised with annotated correspondence labels. The parameters of Arbicon-Net and CNNGeo in this test are trained from scratch except for fixed layers from ResNet-101. We repeat experiment three times. Each time we select one-fold as testing set and the other folds as training set. In this test, we use regular 2D-Conv to replace 4D-Conv in Arbicon-Net for fairness.

**Results Analysis:** Table 3 lists the PCK scores and $E_{smooth}$ as the comparison results between Arbicon-Net and CNNGeo with different geometric model settings. Among these models, our Arbicon-Net achieves better estimation of geometric transformation based on a higher PCK score. This suggests that our Arbicon-Net is more suitable for the estimation of geometric transformation for image pairs in real applications. In addition, Arbicon-Net produces even smoother displacement function with smaller $E_{smooth}$ value compared with non-rigid geometric transformation model TPS. Fig.8 shows two pairs of images pre- and post-registration (warping). We can clearly see from the figure that Arbicon-Net is able to smoothly transform the source image to the target one without introducing significant image distortion, as shown in third column. In contrast, the CNNGeo significantly deteriorate the source image after warping. This further validates that Arbicon-Net can predict a spatially continuous and smooth displacement field as one of our key contributions in this paper.

## 5   Conclusion

We present a novel Arbicon-Net for image registration, which directly learns a dense displacement field between input image pairs. Compared with hand-crafted geometric models, our network is more capable in modeling arbitrary high dimension transformation function. The network structure preserves the predicted displacement function to be spatial continuous and smooth and thus removes the limitation of adding penalization term in training. The experiments in semantic alignment task demonstrate the effectiveness of our approach.

## 6   Acknowledgement

We would like to thank the reviewers for their thoughtful comments and efforts towards improving our manuscript. This work is partially supported by ADEK Grant (No. AARE-18150).

## Footnotes

*Equal contribution to this paper

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
