[Reviews · NeurIPS 2019]

Reviewer 1



The paper presents a neural network model for image registration, which generates an arbitrary displacement field to transform the input image in a way that matches the target. This neural network has several components, including a common feature extraction model that results in a 4D tensor with the correlations of local features from both images. The tensor is then transformed into a vector representation of the transformation, and later used to reconstruct a displacement field. COMMENTS Overall, the work is relatively well presented and provides details to understand most of the formulation and solution. However, there are some confusing aspects that could be clarified or stated more prominently. * My understanding is that the components described in section 3.2 and 3.3 are the central contribution of this work. Section 3.1 describes a strategy used before by other researchers, as well as the loss functions, which seem to be standard and adapted for this work. Is this correct? * I found it difficult to understand the motivations behind these two components. While it seems reasonable to use them and the design looks coherent, no much discussion is provided about why the authors think this is the way of modeling the architecture. * The number of 4D conv-layers is not mentioned, so apparently it's only one layer. How critical is the 4D convolution in this architecture? * The geometry of the filters (Sec. 4.1) does not match 4 dimensions: I assume a tensor with dimensions (w,h,w,h), while the size of the kernels is (3,3,3) with channels (10,10,1). Can you clarify? * I understand another interesting component of the proposed network is the displacement field predictor, which replicates the transformation vector n times, with n, the number of 2D points in the displacement field. I could not follow the continuity argument completely, and the smoothness vs complexity either. The authors say that they prove that spatial continuity is guaranteed, but the provided explanations don't seem to be a sufficient proof to me, unless I missed something. * The experimental results seem generally coherent, but the terminology does not quite match the conventions in the solution. More specifically, parametric and non-parametric transformations is something that was not mentioned in the formulation, and it is difficult for the reader to follow exactly what the authors mean. * The writing style needs polishing. In general, the ideas are well organized, but the text needs grammar improvements all around. The current version is distracting and makes it difficult to follow some details. In summary, the paper has interesting ideas, but it still needs to improve quality of presentation significantly to be a robust submission.

Reviewer 2



Originality I cannot comment much about the originality of this work because I am not familiar with the related works of this research field. To my best understanding, the image encoder, section 3.1 and 3.2 until right before the global descriptor MLP, is proposed by [13]. Putting aside the image encoder, which can be easily plugged in with any better one, I believe the arbitrary continuous displacement field predictor by itself along with the smoothness proof may be a significant contribution that the proposed parameterization of deformation field that is free from any geometric constraint performs better than the previous works with geometric constraints. The baseline methods in experiments are modified to use the same image encoder for fair comparison which strengthens the contribution of this paper. Quality I believe the proposed method is sound and well-motivated. This is a self contained paper with interesting core idea with clear desirable properties. The results are convincing and the baseline seems to be relatively recent works with improvements for fair comparison. Yet, inconsistency in baseline is making the comparison of performance difficult. I believe it would be helpful to report CNNGeo-4D result on experiment 4.3 and use at least the best performing baseline for experiment 4.4 and 4.5. Clarity The paper is well-written and easy to follow. It clearly motivates the problem and explains the details of the network in a degree that it is reproducible. The experiment protocol and the dataset are clearly described and convincing. Minor comment: typo does => does line 148 Significance Although I am not familiar with this field, I find this paper interesting. For previous works, it makes perfect sense to apply pre-defined parameterization of known geometric transformation to model deformation, or to apply additional constraint for smoothness, based on traditional computer vision and graphics knowledge. However, the method this paper is proposing gives me an interesting insight that we instead can train a single continuous function to output a smooth and continuous displacement field without any geometric assumption. From my 3D vision background, this lesson aligns with recent trend of 3D parameterization in a single continuous function such as DeepSDF or OccupancyNet. The paper seems to try to make a fair comparison against recent related works and clearly demonstrates the strength of the proposed method.

Reviewer 3



This paper takes a reasonable approach the task, and the results do appear competitive with state of the art methods. The authors must clarify which parts of their approach constitute novel contributions as opposed to existing technology. In particular, Section 3.1, appears to contain the same content as Section 3.1 in Neighborhood Consensus Networks by Rocco et al.. Equation (1) is copied verbatim. While this work is cited as [13] in other sections of the submission, it is **not** mentioned in Section 3.1. A citation as well as discussion must be added. In addition, I would like the authors to clarify this point and make clear their contributions. The proof that the learned displacement field is smooth needs requires some clarification. While it defers to the universal approximation theorem and regularization techniques, these claims do not constitute a proof. Additional discussion and empirical results would be necessary to support the smoothness claim. The authors provide only a few qualitative results and do not discuss failure cases. This would be necessary to better evaluate the work. Minor comments: 13: "is prov*en*" 112: "further *amplify*" 118, 129: commas should be on the same line as the euqations 120-121: some more intuition on the normalization of the correlation tensor would be helpful 147: "trivially prove*n*" 148: "dose" --> "does" 217: "correspondence*s*"

[Author Response · NeurIPS 2019]

We are thrilled that all reviewers are supportive of our work. We addressed the valuable comments raised below.

**Q1: Clarification of our contribution. (To Reviewer 1, 2 & 3)**

**1.** Please kindly allow us to clarify our key contribution using reviewers' comments as follows:"*For previous works, it makes perfect sense to apply pre-defined parameterization of known geometric transformation to model deformation, or to apply additional constraint for smoothness, based on traditional computer vision and graphics knowledge. However, the method this paper is proposing gives me an interesting insight that we instead can train a single continuous function to output a smooth and continuous displacement field without any geometric assumption. From my 3D vision background, this lesson aligns with the recent trend of 3D parameterization in a single continuous function such as DeepSDF or OccupancyNet.*".

**2.** Our proposed Arbicon-Net is the first original work for 2D parameterization of arbitrary displacement field using a single continuous function in the form of trainable multilayer perceptron (MLP) for image registration. More specifically, our main contribution is explained in Section 3.3, where we clearly present how the single continuous function is implemented with MLP, which outputs a smooth and continuous displacement field without any geometric assumption and smooth regularization.

**3.** Thanks to reviewers' pointer, we are excited to learn about the alignment with the latest trend of 3D surface parameterization (e.g. DeepSDF, CVPR'19, June, 2019) in a single continuous function, which further verifies the technical soundness of our proposed image registration approach. We will cite the related 3D vision references in the revised version.

**4.** We regret that we overlooked the presentation of our idea in an effective manner, leading to reviewers' confusion in understanding our novel contributions. We will revise our paper according to reviewers' valuable comments to clearly present and highlight the key contributions of our work and enhance the proof and the discussion of continuity and smoothness in its final version.

**Q2: Clarification of the design of the components. (To Reviewer 1 & 3)**

**1.** In section 3.3, we motivate to design a trainable continuous function to output a smooth displacement field. We choose MLP as the function because it in theory can learn a fully continuous function with arbitrary precision according to the universal function approximation by neural networks. We designed the "replicate and concatenate" operator because it can largely ensure the smoothness of the estimated displacement field as the function output (Please kindly refer to **Q3** for the justification of this design of the operator that led to the smoothness). **2.** In section 3.2 of transformation descriptor encoder, we leverage the power of 4D convolutional for image correlation feature learning by integrating additional neighborhood information. Note that the 4D convolutional was originally introduced in [13 of our paper], not in our paper. In revised version, we will place related references properly to clarify what are our original contributions and what are others' works. **3.** Kindly refer to experimental section 4.2 where we verified the effectiveness of the 4D-Conv by comparison between *WeakAlign* and *WeakAlign-4D* .

**Q3: Clarification of Continuity & Smoothness. (To Reviewer 1 & 3)**

**1.** Our MLP based displacement function is a continuous function that, for a given 2D spatial point on image plane, outputs the point's displacement values. However, since a MLP often takes as input high dimensional features, it suffers from the "curse of dimensionality" (i.e. when the dimensionality increases, the volume of the space increases so fast that the available data become sparse), which consequently leads to the over-fitting causing the unstable oscillation of the learned function. Current works usually [1 of our paper] impose an additional smooth regularizer (i.e. a linear operator on spatial gradients of the function) to penalizes local spatial oscillation in the function. In contrast, our proposed "replicate and concatenate" operator is designed to help address this challenging issue as explained below: our MLP takes as input a high dimensional feature which is formed by concatenating a 2D point coordinate and the shared common transformation descriptor. Since the shared common transformation descriptor is constant for all 2D point, our MLP is essentially still defined on 2D even though taking input as the high-dimensional feature, therefore the available data wont become sparse in our case. Our strategy for the avoidance of curse of dimensionality contributes to a continuous and smooth function without imposing any regularizer. In addition, as discussed in section 3.3, we also adopted training strategies (e.g. dropout, weight decay) to prevent our deep neural network from over-fitting, which naturally help our Displacement Field Predictor network reduce the risk of the oscillatory of displacement function. **2.** Please kindly refer our respected reviewers to the latest DeepSDF (Jeong, etc, CVPR'19, June,2019, pointed out by reviewer 2) for more detailed discussions on the continuity and smoothness of the single function used for 3D smooth surface parameterization. **3.** As suggested by reviewer 3, in the revised version we will include more empirical results about the smoothness property as well as additional description on the continuity and smoothness explained above.

**Q4: Other questions. (For Reviewer 1, 2 & 3)**

**To Reviewer 1:** The number of 4D-Conv layers is three, where the kernel size we used is $3 \times 3 \times 3 \times 3$. We define "parametric" as pre-defined parameterization of known geometric transformation to model deformation, such as examples in L219. In contrast, "non-parametric" is referred to a displacement field without any geometric assumption. We will clarify these questions in final revised version. **To Reviewer 2:** We will add the comparison result between our method and CNNGeo-4D, which is the best performing baseline on experiment 4.3, on experiment 4.5. **To Reviewer 3:** We will add the missing citations in section 3.1 and more qualitative results and discussion about failure cases in the revised version.We will correct all typos and minors pointed out from all reviewers in revised version.

[Meta-Review · NeurIPS 2019]

This submission received mixed ratings. The most positive reviewers has a non confident rating. R1 and R2 appreciate that the paper is well written and presents an interesting approach to image registration. R1 and R3 point out that the central contribution is not clearly stated in the text. Also overlap of text in sections 3.1-3.3 with previous work exists. In the discussion R3 argues that the paper combines previous work in 3.1/3.2 which limits the novelty and in addition that more clarification and extensive testing is required. R1 agrees that the theoretical section is rather confusing. The authors agree that there is overlap in Section 3.3 with DenseSDF and Occupancy Networks which they rate as not problematic as CVPR19 release was after the NeurIPS submission deadline. Most reviewers agree that there is value to the the combined method presented but exposition needs to be improved. With confidence in that the authors will do as promised in the rebuttal and revise the entire Section 3 and add references therein the agreement was to accept this submission. The novelty part of the submission are sufficient, but the presentation of the paper needs to be improved along the review remarks.